# Role of Functional MRI in Liver SBRT: Current Use and Future Directions

**DOI:** 10.3390/cancers14235860

**Published:** 2022-11-28

**Authors:** Sirisha Tadimalla, Wei Wang, Annette Haworth

**Affiliations:** 1Institute of Medical Physics, School of Physics, Faculty of Science, The University of Sydney, Camperdown, NSW 2006, Australia; 2Crown Princess Mary Cancer Centre, Sydney West Radiation Oncology Network, Western Sydney Local Health District, Sydney, NSW 2145, Australia

**Keywords:** magnetic resonance imaging, MRI, liver, SBRT, SABR, radiation therapy

## Abstract

**Simple Summary:**

The use of magnetic resonance imaging (MRI) for liver radiation treatment planning, adaptation and response assessment is an active area of clinical research. Particularly, research on the use of functional MRI to optimise radiation dose distribution and perform mid-treatment adaptation has grown in recent times. The aim of this review is to provide the current state of evidence on the most relevant functional MRI methods being investigated for liver radiation therapy applications aimed at the development of individualised, adaptive treatments for patients with liver cancer.

**Abstract:**

Stereotactic body radiation therapy (SBRT) is an emerging treatment for liver cancers whereby large doses of radiation can be delivered precisely to target lesions in 3–5 fractions. The target dose is limited by the dose that can be safely delivered to the non-tumour liver, which depends on the baseline liver functional reserve. Current liver SBRT guidelines assume uniform liver function in the non-tumour liver. However, the assumption of uniform liver function is false in liver disease due to the presence of cirrhosis, damage due to previous chemo- or ablative therapies or irradiation, and fatty liver disease. Anatomical information from magnetic resonance imaging (MRI) is increasingly being used for SBRT planning. While its current use is limited to the identification of target location and size, functional MRI techniques also offer the ability to quantify and spatially map liver tissue microstructure and function. This review summarises and discusses the advantages offered by functional MRI methods for SBRT treatment planning and the potential for adaptive SBRT workflows.

## 1. Introduction

Stereotactic body radiation therapy (SBRT) is increasingly utilised for treatment of primary liver cancer as well as liver metastases, most often in patients who are not eligible for other local therapies such as surgical resection, ablation or embolization techniques [1,2,3]. SBRT uses several, precisely focused beams of radiation, enabling the delivery of higher radiation dose to the tumour with less dose to surrounding liver parenchyma in fewer fractions than other external beam radiation treatment techniques. High-dose SBRT for unresectable primary and metastatic liver cancer has demonstrated excellent local control rates [4]. However, the ability to deliver higher doses safely is limited by the risk of toxicity in the non-tumour liver. A critical volume of 700 cc, receiving <15 Gy in 3 fractions is recommended for SBRT for liver metastases, with the volume estimate adapted from partial hepatectomy guidelines [5]. In the case of primary liver cancer, such as hepatocellular carcinoma (HCC) where liver function is already compromised, the goal is to deliver a therapeutic dose of radiation whilst minimising the risk of deterioration of liver function.

Traditionally, normal tissue complication probability (NTCP) models that predict liver toxicity are based on dose-volume parameters [6]. However, these models do not take into account the heterogeneity of an individual patient’s liver function prior to treatment, or the heterogeneity of changes in local liver function as a response to radiation as treatment progresses. Efforts are ongoing on the use of functional imaging techniques to obtain the spatial distribution of liver function to guide treatment planning for selection of poorly functioning liver regions that can be sacrificed in order to spare more high functioning liver [7]. These methods may also be useful for assessing early local changes in liver function to radiation treatment, which can support personalised treatment adaptations such as mid-treatment dose escalation to maximise tumour control, or for identifying patients at greater risk for radiation-induced liver disease (RILD).

In this review, we provide an overview of the various methods used for liver function assessment, focusing on imaging techniques for the estimation of regional liver function. We will then introduce magnetic resonance imaging (MRI) techniques for functional liver imaging and their current application in liver SBRT. Finally, we will discuss the challenges, limitations and potential future directions for functional MRI in liver SBRT.

## 2. Liver Function Assessment

### 2.1. Global Liver Function Tests

Liver function is currently assessed via scoring systems such as the Child-Pugh (CP), Albumin-Bilirubin (ALBI) and Model for End-stage Liver Disease (MELD) scores, all of which are based on biochemical markers indirectly linked to liver function [8,9]. Liver SBRT dose prescriptions for HCC are commonly based on the CP scoring system [10]. CP scores can range from 5 to 15 and are classified as CP-A (5–6 points, well-compensated cirrhosis), CP-B (7–9 points, cirrhosis with significant liver function compromise) and CP-C (10–15 points, decompensated cirrhosis). The CP class and score reflects hepatic functional reserve and serves as a prognostic factor in patients with HCC. SBRT is considered relatively safe and effective in patients with CP-A scores, while increased risk of radiation-induced toxicity and poor treatment outcomes post-SBRT have been reported in patients with CP-B7 and CP-C scores [1,11]. While there are reports of patients with CP-B7/8 scores safely treated with SBRT, dose prescriptions are frequently lower and stricter dose constraints are applied to non-tumour liver compared to dose prescription in patients with well-compensated cirrhosis [1,12]. For example, the QUANTEC report on radiation associated liver injury recommends significant lower mean liver dose of less than 6 Gy in the context of SBRT for primary liver cancer in a patient with CP-B cirrhosis [5]. However, the global liver function scoring system classifies patients into broad categories of liver function and RILD risk, and does not allow for individualised treatment approaches.

The Indocyanine Green (ICG) test is a biochemical test, where the retention of the intravenously administered fluorescent ICG dye at 15 min (ICG-R15) provides a more direct measure of overall liver function. The ICG is cleared from the bloodstream exclusively by the liver, and if more than 15% of the ICG remains in the circulation 15 min after injection, then liver function is considered to be impaired. ICG tests are routinely performed to assess liver function in patients after liver transplantation or to assess functional liver reserve prior to surgical liver resection [13,14]. ICG tests have also been used to predict liver function following SBRT in patients with HCC [15]. ICG-R15 has been correlated with RILD and adaptive normal tissue response models based on estimations of functional liver reserve using ICG tests have been developed [16]. These models offer a pathway for the development of personalised treatment approaches. Mid-treatment adaptation of liver SBRT in patients with HCC and liver metastases based on liver function assessment using ICG tests has been demonstrated in a Phase II clinical trial [17]. However, ICG tests can only provide estimates of overall liver function, and cannot be used to map the spatial variation in liver function seen in patients with liver cancers.

### 2.2. Imaging of Liver Function

Computed tomography (CT) techniques such as dynamic contrast-enhanced computed tomography (DCE-CT) with pharmacokinetic modeling have shown that portal venous perfusion, described by the portal venous blood flow, correlates with overall liver ICG-R15 measurements, which also reflect liver perfusion. Reduction in portal venous perfusion has been investigated as a measure of liver function dose response, given that RILD is characterised by the development of venous occlusions. Furthermore, regional portal venous plasma flow is correlated with local radiation dose and is a predictor of post-treatment radiation-induced liver dysfunction [18,19,20]. As liver fibrosis is another likely consequence of irradiation, dual energy CT (DECT) after iodine contrast injection has been used to map iodine density across the liver, reflective of liver fibrosis [21]. CT-based methods are inexpensive, quick and accessible, and in the case of DCE-CT have the advantage of a linear relationship between image intensity in Hounsfield units and tracer concentration. However, they generally have the disadvantage of increased radiation exposure and the need for injected iodinated contrast agents. Furthermore, DCE-CT can only provide information on liver perfusion, but not on hepatocellular function.

Various single-photon emission computed tomography (SPECT) ${}^{99m}$Technicium (Tc) tagged tracers, using sulphur colloid (SC), hepatobiliary iminodiacetic acid (HIDA) or galactosyl human serum albumin (GSA), have also been used to assess liver function [22]. The radioactive tracer ${}^{99m}$Tc-SC is taken up by Kupffer cells of the hepatic reticulo-endothelial system, and the uptake has been used to identify regions of high functioning liver [23]. SPECT image parameters such as threshold-based functional liver volumes (FLV) relative to anatomical liver volume, mean liver-to-spleen uptake ratio (LSR), and total liver function (TLF) ratio derived from the product of FLV and LSR have been correlated with CP classification and albumin and bilirubin levels, and are also predictive of RILD [24,25]. ${}^{99m}$Tc-GSA binds specifically to the asialoglycoprotein receptor (ASGPR) expressed on hepatocytes, and the uptake reflects the density of the ASGPR, indirectly relating to liver function. Parameters such as the liver uptake value and functional liver index in ${}^{99m}$Tc-GSA SPECT images have been correlated with liver fibrosis scores [26]. ${}^{99m}$Tc-HIDA is transported via albumin by hepatocyte membrane protein transporters and excreted into the biliary system. Low uptake of the tracer reflects reduced activity of the transporters, indicating impaired liver function [22]. Positron emission tomography (PET)-CT imaging of liver function using ${}^{18}$flourine (F) tagged fluoro-D-galactose (FDGal, a galactose analog metabolised by hepatic cells) has also been used to quantify hepatic metabolic function [27,28]. Tracer kinetic modeling of the dynamic signal has been used to also derive quantitative tissue perfusion parameters such as hepatic arterial or portal venous blood flow and volume [29]. A clinical trial is underway at the Princess Alexandria Hospital in Australia to evaluate the use of ${}^{18}$F-FDGal PET for liver function assessment and to predict toxicity of liver SBRT (https://www.australianclinicaltrials.gov.au/anzctr/trial/ACTRN12619001499178, accessed on 27 September 2022). However, the limited availability of specialist and expensive nuclear medicine facilities is a downside of SPECT and PET based methods.

On the other hand, MRI is widely available and most patients with liver cancers already undergo several MRI scans as part of diagnostic, treatment and follow-up procedures. MRI offers high spatial resolution and full liver coverage, with the ability to acquire several functional images giving complementary information on tissue perfusion, cellular function and microstructure in a single imaging session. In the next section, we will discuss some of the functional MRI techniques that are used to assess liver function.

## 3. Functional MRI of the Liver

A typical liver MRI protocol consists of T2-weighted (T2w) imaging, in- and opposed-phase (IP/OP) T1-weighted (T1w) imaging, T2*-weighted (T2*w) imaging, diffusion-weighted imaging (DWI) and dynamic contrast-enhanced MRI (DCE-MRI) [30]. The anatomical T2w images provide excellent high-resolution definition of most anatomical structures in the abdomen, enabling accurate delineation of the gross tumour volume (GTV) and organs at risk (OAR). They also allow for precise measurements of liver volume, both for treatment dose calculations as well as the assessment of treatment response of the tumour and the radiation response of the non-tumour liver. IP/OP images are used to separate fat and water signal components and provide complementary anatomical detail for more precise localisation of structures. Quantification of liver fat fraction from the IP/OP images is typically used for diagnosis of fatty liver disease, and has limited use in current SBRT planning or response assessment. Similarly, T2*w imaging is typically used to quantify liver iron content for the diagnosis of iron overload, but has limited value in the context of liver SBRT planning and response assessment.

### 3.1. Diffusion-Weighted Imaging

DWI is a functional MRI technique which derives image contrast from the differences in the amount of diffusion of water molecules in different tissues. Diffusion is a physical process of random, microscopic movements of water molecules due to their thermal energy (Brownian motion). As tissues with different micro-structures, cell densities, vascular networks, and composition present different levels of barriers to water molecular diffusion, DWI can provide an indirect assessment of tissue characteristics such as cellularity, micro-circulatory perfusion and fibrosis. DWI is being increasingly applied for improved confidence in lesion detection and characterisation as well as in predicting and monitoring tumour response to treatment [31]. HCC lesions, which are highly cellular and vascular, show increased restriction to diffusion, and appear hyperintense in DWI images [32]. DWI has also been used in detection of metastatic colorectal, pancreatic and neuroendocrine lesions in the liver, and when combined with contrast-enhanced MRI, the sensitivity of detection is more than 90% [33,34,35].

DWI is performed using strong gradients that sensitise the MRI signal to the diffusion. The sensitivity of the signal to diffusion is controlled by varying the strength and timing of the gradients (the b-value) during imaging. Several DWI images are acquired at different b-values, commonly ranging from b = 0 to 800 s/mm^2^ or more. A quantitative diffusion parameter, the apparent diffusion coefficient (ADC), derived via mono-exponential modeling of the DWI signal at b = 0 s/mm^2^ and two or more b-values, is widely used to evaluate liver tumours. Malignant liver tumours have a reduced ADC compared to benign tumours and healthy liver tissue, although the values can overlap with the ADC in other solid benign lesions such as hemangioma and hepatocellular adenoma [36]. An increase in ADC is associated with positive treatment response to various loco-regional therapies of hepatic cancers due to treatment-induced necrosis [37]. Koinuma et al. evaluated ADC measurements in patients with chronic liver disease, with various CP scores. They reported an inverse relationship between ADC and fibrosis scores obtained through a liver biopsy, and also found that liver ADC values were lower in patients with poorer liver function as indicated by their CP scores [38]. Other studies have shown that the ADC is also reduced in the presence of hepatic inflammation and fibrosis in patients with chronic hepatitis [39,40], demonstrating a link between liver ADC values and liver function. Hollingsworth and Lomas examined the influence of liver tissue perfusion on ADC measurements by calculating the ADC using three pairs of b-values (0 and 200 s/mm^2^, 0 and 500 s/mm^2^, and 0 and 750 s/mm^2^) in healthy volunteers before and after a meal [41]. They found that only the ADC calculated with b = 0 and 200 s/mm^2^ reflected the expected post-prandial increase in liver portal venous perfusion. This is because at low b-values, typically less than 200 s/mm^2^, the DWI signal is additionally sensitive to micro-circulatory perfusion.

The intra-vascular incoherent motion (IVIM) model fitted to DWI image series acquired at several (at least 5) b-values with 3 or more less than 200 s/mm^2^ can be used to separate the perfusion and diffusion components of the DWI signal, giving a pseudo-diffusion coefficient (D*) which reflects the rate of tissue perfusion and a perfusion fraction (f) which reflects the amount of perfusion, in addition to the diffusion coefficient D [42]. Although in principle, both D* and f describe the perfusion component of the signal, only the D* parameter has shown statistically significant correlation with post-prandial portal venous flow measurements obtained using phase contrast imaging [43]. However, IVIM has been used to evaluate the hepatic functional reserve in liver cirrhosis; D*, f and D are all significantly lower in cirrhotic livers and decrease with increasing CP scores [44].

Both the mono-exponential and IVIM models rely on the assumption that diffusion is gaussian, which is false in biological tissues. The deviation from the gaussian model is more apparent at higher b-values (>1000 s/mm^2^) and is described by the kurtosis of the distribution, which is incorporated into the Kurtosis model of diffusion [45]. Several DWI images with at least one obtained at a very high b-value (>1500 s/mm^2^) are fitted to the Kurtosis model, to calculate diffusion kurtosis K_app_ along with a corrected diffusion coefficient D_K_. It is recognised that ADC values calculated at b-values below 1000 s/mm^2^ primarily assess diffusion in the extracellular space. In contrast, K_app_ and D_K_ additionally describe the interactions of water molecules with cell membranes and intracellular structures [46]. Mean liver K_app_ correlates moderately with CP score and ICG-R15 measurements [47,48], and was found to be higher in patients with HCC whose liver function was found to deteriorate (change in CP class from B to C) following trans-arterial embolization treatment, compared to those who did not show a change in CP score [49]. Impaired hepatocellular function is linked to hepatic fibrosis accompanied by swelling of hepatocytes and changes in composition of the extracellular space. As K_app_ reflects the heterogeneity of tissue microstructure, diffusion kurtosis imaging (DKI) shows promise for functional liver imaging. A summary of the various DWI models and their parameters is given in Table 1.

### 3.2. Dynamic Contrast-Enhanced MRI

Dynamic contrast-enhanced MRI (DCE-MRI) is a functional MRI technique that involves the acquisition of several images before and after the intravenous injection of an MRI contrast agent. MRI contrast agents are typically paramagnetic Gadolinium ion ($Gd^{3+}$) complexes which shorten the T1 and T2 relaxation times of nearby water protons, and in effect increase the signal intensity of these regions in T1w images and decrease the signal intensity in T2w images. As the effect on T1w images can be achieved at much lower gadolinium concentrations, in conventional clinical practice, DCE-MRI is performed by evaluating T1w images before and through successive phases of contrast enhancement. Most MRI contrast agents are extracellular, that is, they distribute from the vascular space into the extracellular space, and are excreted by glomerular filtration. The kinetics of these contrast agents provide information on vascular permeability and tissue perfusion. Liver-specific MRI contrast agents, also known as hepato-biliary agents, are intracellular; they are actively taken up from the extracellular space into the hepatocytes by protein membrane transporters. Therefore, these contrast agents have two excretory pathways, via the biliary network as well as through glomerular filtration. The hepatobiliary kinetics provide additional information on hepatic function.

Typically, pre-contrast and multi-phase or pseudo-continuous post-contrast T1w images are acquired over a period of a few minutes when using extracellular contrast agents, or over a period of 20–30 min when using hepatobiliary agents. HCC is hypervascular, and most HCCs show a strong early hyperenhancement in the hepatic arterial phase, followed by a hypointense appearance in the portal venous phase. The use of hepatobiliary contrast agents is especially useful for the detection of atypical HCC lesions that are hypovascular, and therefore, do not show the typical enhancement pattern. These lesions appear hypointense in the hepatobiliary phase, where the healthy liver parenchyma appears brighter due to the higher hepatocyte uptake of the contrast agent indicative of normal hepatic function [50]. The ratio of liver-to-spleen image intensity in the hepatobiliary phase is also sometimes used to assess liver parenchymal function [51]. Liver metastases are also hypovascular, and appear hypointense in the hepatobiliary phase. However, the kinetics of the contrast agent in liver metastases is very different from HCC lesions, and the various types of lesions can be differentiated using semi-quantitative and quantitative image analysis.

Semi-quantitative analysis is based on the calculation of heuristic parameters that describe the T1w MRI signal-time curve. These parameters include (a) the area under the curve (AUC), usually calculated as the the amount of enhancement over a period of time, typically 60 or 90 s (AUC60 or AUC90), (b) the maximum or peak enhancement ratio, which is the maximum relative increase in the T1w signal from baseline, (c) the wash-in slope, which determines the rate of enhancement, (d) the mean transit time (MTT), which describes the average time taken for the blood to perfuse the tissue and relates to the blood volume and flow, and (e) the wash-out slope, which relates to the rate of excretion [51,52]. Semi-quantitative DCE-MRI has been used extensively to assess chronic liver disease and parameters such as the wash-in and wash-out slopes and the AUC have been correlated with liver fibrosis [53,54]. Semi-quantitative parameters are easy to calculate, however, they are difficult to compare across studies as the signal-time curves can vary considerably due to differences in the timing of the post-contrast T1w acquisition, the injection rate, etc. Furthermore, the parameters only describe the curve, and do not relate directly to tissue physiological characteristics. When using hepatobiliary contrast agents, another functional parameter that is often used is the hepatic extraction fraction (HEF), which describes the extraction of the contrast agent from plasma by the hepatocytes [52,55,56]. While the HEF is indicative of hepatocyte function and is correlated with CP scores [57], it cannot differentiate between plasma flow and hepatocyte uptake.

Quantitative DCE-MRI is based on tracer kinetic modeling of the image time series. Standard tracer kinetic models assume a single inlet for the contrast agent from the circulatory system to the tissue vascular space, with passive exchange of the contrast agent between the vascular and extracellular spaces. A two-compartment Extended Tofts model, providing model parameters such as the volume transfer constant K^trans^, is adequate for assessment of tumour perfusion in liver metastases [58]. However, as the liver is supplied by the hepatic artery and the portal vein; dual-inlet models are required for assessment of HCC and liver parenchyma. Dual-inlet single compartment model (Figure 1a) parameters include the arterial and portal venous inflow rates, analogous to K^trans^, and the outflow rate constant. The arterial flow fraction, which describes the proportion of supply from the hepatic artery, is derived as the ratio of the arterial inflow rate to the total inflow rate (sum of arterial and venous inflows). Often, two additional parameters, the arterial and venous delays, describing the delay in arrival of the contrast agent from the larger hepatic artery and portal vein to the parenchyma are also included in the model [59]. This model has been extensively used to assess and stage liver fibrosis, with increased arterial flow and plasma volume associated with increased liver fibrosis [54,60].

Pharmacokinetic models that describe the kinetics of hepatobiliary contrast agents are necessarily more complex than perfusion-only models. These models are typically dual-inlet, two compartment models (Figure 1b), and include an additional transfer constant, the hepatocyte uptake rate, which describes the extraction of the contrast agent from the extracellular space into the hepatocytes, and an additional distribution volume, the hepatocyte volume fraction. Optionally, the biliary efflux rate of the contrast agent excretion can also be fitted [61,62]. However, the increased model complexity can lead to reduced precision in parameter estimates due to overfitting and inadequate temporal resolution [63]. Simpler models have been derived utilising fewer model parameters, such as the dual-inlet Patlak model [64] and the linearized two-compartment model [65]. The hepatocyte uptake rate is the main descriptor of liver function and has been correlated with CP score in patients with cirrhosis as well as ICG-R15 measurements and GSA uptake in SPECT images [64,66]. However, for patients with poor liver function, the product of uptake rate and liver volume was found to be better correlated with ICG-R15 and ALBI scores than uptake rate alone [67].

### 3.3. Advanced and Non-Standard Imaging Techniques

Other specialised or advanced functional techniques include magnetic resonance elastography (MRE) which is used to measure liver stiffness for assessment of liver fibrosis and cirrhosis, and arterial spin labeling (ASL) which is an alternative perfusion imaging technique. MRE uses an external vibration generator to measure the elastic properties of the liver tissue, which have been correlated with liver functional reserve measured using ICG tests [68]. In ASL, magnetically labelled water protons in blood are used as an endogenous tracer. By using carefully placed labeling planes, arterial and portal venous perfusion can both be assessed [69]. Although ASL has been used for perfusion measurements in the brain since the 1990s, it is an emerging method for liver perfusion imaging [70,71]. Furthermore, while MRE is an established technique, it requires significant investment in specialist equipment that is not routinely available in all MRI centres. In contrast, DWI and DCE-MRI are widely available and already used in clinical liver MRI and will be the focus of this review.

## 4. Current Use of Functional MRI for Liver SBRT

### 4.1. Use in Treatment Planning

MRI offers excellent soft-tissue resolution for the detection and characterisation of liver tumours. While CT remains the workhorse of radiation therapy, MRI has several advantages such as zero ionising radiation exposure, superior tissue contrast, availability of hepatocyte-specific contrast agents and the ability to obtain anatomic and functional images within a single scan. MRI is, therefore, increasingly being used to provide complementary information to CT images for SBRT planning, although it is not mandated in most liver SBRT guidelines [72,73,74]. In current liver SBRT treatment planning, MRI is often fused with CT to guide target delineation. However, this requires accurate image registration between CT and MRI, which adds to the complexity of liver SBRT planning. MRI-only planning has already been developed in brain, head and neck and prostate, and is an emerging alternative in other body sites [75]. Using deep learning, Liu et al. have developed methods to generate synthetic liver CT images from 3D T1w MRI images that achieved SBRT dose distributions comparable to a traditional CT image-based plan [76]. A major concern with the use of MRI for SBRT treatment planning is the presence of geometric distortions in MRI images, which can lead to dosimetric uncertainties. In a simulation study [77], it was found that distortion of up to 3 mm, measured separately on phantoms and averaged across several T1w and T2w MRI sequences, result in small dose uncertainties of <1 Gy. However, for small targets ( 15 mm diameter), the impact can be substantially larger, resulting in dose errors of up to 15% [78]. Breathing motion is another source of error in treatment planning, as traditional MRI image acquisition is relatively slow and several breathing cycles pass by the time an image is acquired, leading to image blurring. Respiratory-triggered image acquisition with the patient in the treatment position using optimised MRI sequences can improve image quality although there is the penalty of increased scan times [79]. Implantation of radio-opaque fiducial markers to guide treatment delivery and real-time tumour tracking using external intra-fraction monitoring devices are other approaches to motion management [80,81]. Additionally, emerging methods of 4D MRI and hybrid MR-linac (MRI + linear accelerator) can track the motion of tumours and other structures in real-time, enabling adaptive treatment and significantly reducing the impact of breathing motion on treatment accuracy [82,83,84]. While the use of functional MRI in this context is limited, there is growing evidence to suggest that gadoxetate enhancement can allow the sustained visualisation of tumours that have poor visibility on anatomical MRI [85,86]. However, repeated administration of Gadolinium-based contrast agents has been linked to Gadolinium deposition in the brain [87], therefore, more research on low-dose gadoxetate MRI as well as other functional MRI techniques such as DWI is required. While currently not used in SBRT planning, preliminary studies have shown that DWI can improve accuracy in tumour delineation [88]. Future implementations of liver SBRT on MR-linacs can include functional MRI methods to enable daily personalised treatment planning and optimisation of SBRT dose.

Regional liver function measurements are not currently used to guide liver SBRT planning. While research in this area is still in its infancy, preliminary studies have showed promise. Proof of concept studies using retrospective data from small groups of patients (<20) with HCC and liver metastases have demonstrated benefits of using spatial liver function maps to identify and spare high functioning liver from irradiation in terms of significant dose reduction to functional liver, while maintaining target coverage and OAR sparing comparable to plans designed based on the anatomy only [21,23,89,90,91,92,93]. A prospective clinical trial on 15 participants has been recently performed to compare differences in the functional liver reserve when using SBRT plans based on SPECT-HIDA scans vs. standard SBRT plans. The trial also evaluated the proportion of patients for whom the SPECT-based SBRT plans were chosen for treatment over the standard SBRT plans (NCT03338062: A pilot study to assess theragnostically planned liver radiation to optimize radiation therapy). Preliminary results show that 4 of the 15 patients showed >5% improvement in functional liver reserve, and the SPECT-based plans were selected for 11 patients [94]. Larger prospective trials with longer follow up are needed to determine the safety and efficacy of the function-based approach.

In studies conducted thus far, liver function was measured semi-quantitatively—as a percentage of maximum tracer uptake in SPECT images, normalised iodine density (NID; obtained as the ratio of iodine density in liver and aorta) in DECT images, and liver-to-spleen intensity ratios in gadoxetate DCE-MRI images. Only one study used pharmacokinetic modeling of gadoxetate DCE-MRI to derive quantitative perfusion and uptake rates [93]. In studies using SPECT, high functioning liver was identified based on thresholds determined from published correlations between the parameters and clinical liver function measures such as CP scores [95,96], and was defined as regions with tracer uptake ≥ 50% relative to the maximum uptake. For quantitative perfusion and uptake rates derived from gadoxetate DCE-MRI, voxels with uptake and perfusion >36% of the values measured in a manually selected region with ’normal’ function were defined as high functioning [93]. There is as yet no global consensus on the definition of high functioning liver in imaging studies. Furthermore, as most of the liver function estimates are not direct measures of hepatocellular function, there can be considerable variation in the function-based plans. For example, although liver perfusion and function are closely related [97], Simeth et al. have reported that mismatch in definitions of high functioning regions based on perfusion and uptake spatial maps can result in variations as large as 10% in the mean dose reduction to functional liver [93]. Wei et al. have similarly found that the dose that causes a 50% loss in function (gadoxetate uptake rates) is significantly lower compared to the dose that causes the same loss in perfusion (portal venous flow) [98]. The relationship between liver perfusion and function was also found to vary between patients with HCC tumours and patients with non-HCC tumours (liver metastases and cholagiocarcinoma), with the same amount of liver perfusion translating to less probability of liver function in patients with HCC [99].

Despite correlations observed between chronic liver disease and DWI-derived parameters, there have been no studies using functional DWI parameters to guide liver SBRT planning, and the role of this functional MRI technique in liver SBRT planning remains unexplored.

### 4.2. Use in Response Assessment

In the clinic, response assessment after liver SBRT follows general assessment criteria for liver cancers, such as the Response Evaluation Criteria In Solid Tumours (RECIST), modified RECIST (mRECIST), the European Association of Study of the Liver (EASL) or the Liver Imaging Reporting and Data System (LI-RADS) criteria. These criteria typically assess changes in tumour size and in the case of mRECIST, EASL and LI-RADS criteria, also include observations on arterial enhancement of the tumours in contrast-enhanced CT or MRI images. Imaging is usually performed every 3 months after treatment for response evaluation. After SBRT, HCC tumours shrink slowly, however, change in size is not always correlated with treatment success [100]. Price et al. suggest that reduced vascularity of HCC tumours or non-enhancement in DCE-CT/DCE-MRI indicating necrosis is potentially a more useful indicator of treatment success than tumour size [101]. However, the presence of persistent arterial phase hyperenhancement, even up to 1 year following SBRT, does not necessarily indicate residual viable tumour [102,103]. The RECIST criteria is similarly limited in the response evaluation of liver metastases following SBRT, and criteria such as mRECIST and EASL which combine assessment of size as well as contrast enhancement are recommended [104]. However, there is no general consensus on the optimal criteria to use for response assessment after liver SBRT [104]. Furthermore, all the response assessment criteria are qualitative, and functional, quantitative liver MRI is rarely, if ever, used clinically.

There are no assessment criteria that account for the radiation-related response of the liver parenchyma. However, several imaging studies report morphological and physiological changes in the immediate surrounding liver parenchyma following SBRT [105]. These changes are collectively referred to as focal liver reaction (FLR). Morphological alterations in the appearance of liver surrounding tumours in T2w and T1w (pre- and post-Gadolinium contrast injection) images have been observed as early as 6 weeks following SBRT [106]. Irradiated liver parenchyma appears hypointense on T1w images, and hyperintense on T2w images, which may be related to hepatic edema which can occur very early after SBRT and is accompanied by a slight increase in ADC values. On hepatobiliary DCE-MRI images, the FLR appears as a well-demarcated hypo-intense region up to 6 months after treatment. Between 1–3 months after treatment, the surrounding irradiated parenchyma exhibits arterial enhancement in DCE-MRI images, which without subsequent wash-out in the delayed phase indicates a focal liver reaction, differentiating it from residual tumour. Another focal reaction is steatosis, which manifests as decreased signal intensity on opposed-phase and T2w images [105,107,108]. Furthermore, a reduction in liver volume has been observed after SBRT (from 3 to >12 months following treatment), with a wide range of volume reduction from 24% to 60% reported in the literature [105,107]. Fibrosis is another response of liver parenchyma to radiation, and manifests as delayed enhancement on DCE-MRI images [107].

Tumour ADC (in both HCC and liver metastases) has been observed to increase significantly compared to baseline as early as 1 week into treatment. In HCC patients treated with SBRT, a tumour ADC increment within 6 months following treatment has been observed to be indicative of favorable response, with estimated minimum effect size ranging from 20% to 25% [109,110]. However, these studies were retrospective, included small numbers of patients with short follow up times and only assessed tumour response radiologically, with no histological confirmation. Increase in ADC following liver SBRT has also been observed in peri-tumour regions and irradiated liver parenchyma, with no significant change in non-irradiated liver parenchyma [111]. The increase in tumour ADC has been correlated with radiation dose [111]. However, there is no robust evidence for correlation between increase in parenchymal ADC with radiation dose, with some preliminary studies showing that the ADC increase occurs in regions irradiated with low (<15 Gy) and high (>30 Gy) doses [112]. Few studies have evaluated the predictive/prognostic value of IVIM or DKI in the context of liver SBRT. This may be due to the greater precision errors associated with measurement of IVIM parameters, particularly f and D*, when compared to ADC. While preliminary studies have reported changes in IVIM parameters in liver parenchyma following SBRT, demonstrating sensitivity to dose deposition [113], prospective studies with large sample sizes and improved DWI acquisition and post-processing methods are needed.

Qualitative assessment of DCE-MRI images is already part of routinely used evaluation criteria, with early arterial, portal venous and hepatobiliary phase enhancement used as indicators of tumour response and parenchymal reaction to radiation as described above. Correlations of tumour and hepatic response with semi-quantitative and quantitative DCE-MRI parameters have also been reported in some studies. For example, an increased wash-in slope and peak enhancement in HCC tumours as early as two weeks following radiotherapy has been linked to improved local response [114]. Similarly, quantitative portal venous blood flow to intra-hepatic tumours decreases significantly following treatment, and in the non-tumour liver is found to correlate with ICG-R15 measurements before, during and 1 and 2 months after irradiation [115]. Uptake rates of gadoxetate in liver parenchyma were found to be reduced 1 month post-SBRT in patients with HCC, however, it is possible that the decrease is a combined response to both the delivered radiation dose as well as hepatic inflammation [116]. Further validation of treatment-related changes in the DCE-MRI parameters is required in larger patient datasets with longer follow up times.

### 4.3. Use for Dose-Response Assessment and Mid-Treatment Adaptation

RILD is the main complication from liver SBRT and results from radiation induced damage to the liver parenchyma. The reported complication rate for liver SBRT is <5%, when strict dose-volume constraints are applied and patients without cirrhosis or well-compensated cirrhosis are selected for treatment. RILD is seen most often in patients with pre-existing chronic liver disease and is typically defined by an increase in the CP or ALBI score after treatment [107,108]. Current NTCP models are geared toward predicting the risk of RILD. However, they are based on studies which did not differentiate between patients with HCC, with compromised liver function, and patients with liver metastases, with better liver function. The models also do not account for spatial variation of liver function due to the underlying liver disease and liver damage from prior treatments which can increase the risk of RILD after SBRT.

Approaches for treatment planning guided by the spatial distribution of baseline liver function have been already described in a previous section. In most studies, while the radiation dose distribution was modified to spare high functioning liver, dose–response and post-treatment liver function were not considered during plan optimisation. However, Wu et al. found that plans that are designed to spare highly functioning regions of the liver do not necessarily retain the most post-treatment global liver function [117]. The authors suggest that plans where damaging doses are delivered to fewer high functioning liver voxels can retain more liver function post-treatment as long as doses delivered to these voxels are tolerable. They estimated the thresholds for damaging and tolerable doses (>50 Gy-EQD2 and <20 Gy-EQD2 where EQD2 is the equivalent dose in 2 Gy fractions) based on dose–response relationships obtained from a small, retrospective study within their institution. However, dose–response can vary across patients and depend on the baseline liver function [101]. For instance, using visible focal liver reaction on MRI obtained 3–5 months post-treatment, Yadav et al. estimated an upper dose threshold of 35 Gy mean dose to the liver [118]. Another study reported that the dose threshold for liver dysfunction on post-treatment MRI is different for cirrhotic and non-cirrhotic livers (BED2 = 40 Gy in cirrhotic livers and 70 Gy in normal livers) [119]. Ideally, the liver SBRT dose should be guided by the baseline liver function and liver function dose–response for each individual patient. Predictive mathematical models of tumour dynamics and dose response similar to those being developed in the context of lung cancer treatments [120,121] could enable the identification of patients at risk of post-treatment liver dysfunction and facilitate mid-treatment dose adaptation.

Implementation of individualised, mid-treatment adaptation of liver SBRT has been realised using ICG tests for measurement of liver function [15]. In a Phase II clinical trial on 90 patients (23% CP grade B), pre- and mid-treatment liver function measurements of ICG-R15 have been used to modify prescribed dose after 3 SBRT fractions, successfully identifying patients at higher risk for liver dysfunction who could not be treated further and patients who could tolerate further treatment after a 1 month pause and lower doses at subsequent SBRT fractions. The individualised adaptive approach demonstrated in this trial achieved high rates of local control of 99% at 1 year and 95% at 2 years after treatment, as well as significantly altered the predicted loss of liver function [17]. Another trial with 80 patients with CP-B liver disease and HCC also achieved high rates of 1 year local control of 92% using the same individualised adaptive SBRT approach with ICG-R15 liver function measurements [122], demonstrating the benefit of personalised, adaptive treatment especially for patients with liver cirrhosis at baseline. These studies demonstrate the potential for achieving high rates of tumour control without compromising safety for patients who are already at an increased risk of liver dysfunction. At present, no similar studies using functional liver MRI have been performed. Research is ongoing; a University of Michigan clinical trial (https://clinicaltrials.gov/ct2/show/NCT02460835, accessed on 27 September 2022, NCT02460835: A pilot study of individualised adaptive radiation therapy for hepatocellular carcinoma) aims to develop a function-based planning and adaptive SBRT approach using portal venous perfusion maps to estimate regional liver function and is due to complete in 2023. Another clinical trial commencing in 2022 at the University of Sydney (https://www.anzctr.org.au/Trial/Registration/TrialReview.aspx?id=383543&isReview=true, accessed on 27 September 2022, ACTRN12622000371796: Personalised liver stereotactic body radiation therapy using magnetic resonance imaging (PRISM)) is examining the feasibility of using liver function measurements from pharmacokinetic modeling of gadoxetate DCE-MRI for liver SBRT planning and mid-treatment adaptation.

## 5. Discussion

Functional MRI offers the ability to spatially map liver function in patients with liver cancers, enabling optimisation of SBRT dose distributions to preferentially spare high functioning liver regions, without compromising target coverage and OAR dose constraints. Incorporation of dose–response information on functional MRI parameters can also enable adaptive treatment approaches, leading to personalised dose escalation to improve the therapeutic ratio or de-escalation to reduce the risk of treatment-induced toxicity.

Current research has highlighted the promise of functional MRI for personalisation of liver SBRT, however, there is much work to be done. As there is no ’ground-truth’ measurement of liver function, considerable efforts will be required for the validation of functional parameters as biomarkers of liver function. Various studies have used DCE-MRI parameters such as arterial and portal venous perfusion and hepatocyte uptake rates interchangeably to define liver function and validated them against global liver function measures such as ICG-R15 and CP scores. At present, classification of high and low functioning liver regions is based on relative differences in the spatial distribution of these parameters. This is because there is no absolute threshold to separate parameter values indicating high and low liver function. Ideally, the minimum threshold for high liver function should be identified based on the immediate and long-term dose–response of the functional MRI parameter. Previous studies have reported on longitudinal changes in DWI and DCE-MRI parameters in liver cancer tumours following treatments such as chemotherapy, chemo-embolisation and radio-embolisation [123,124,125]. However, liver SBRT is a relatively new treatment and there is insufficient data on the radiation dose response of these parameters in tumours and the non-tumour cirrhosis-affected liver. Future studies designed to acquire functional MRI data longitudinally with long patient follow-up times are needed to establish dose thresholds. Longitudinal data will also enable the development of predictive models of post-treatment outcome based on baseline and mid-treatment measurements of the functional MRI parameters.

For a functional MRI parameter to demonstrate a clear dose–response, it is also essential that it is reproducible. It is well known that DWI and DCE-MRI derived quantitative parameters suffer from a lack of reproducibility. Considerable efforts are underway to reduce variations in image acquisition and post-processing methods to improve the accuracy and precision of these parameters. However, many of the functional MRI parameters described in this review are novel and their technical performance has not yet been well characterised. Therefore, clinical trials aiming to utilise functional MRI in liver SBRT workflows must consider the uncertainty associated with the measurement of each functional parameter. Quality assurance should be performed in multi-centre studies to measure and monitor parameter uncertainties throughout the period of the clinical trial and establish performance benchmarks.

Finally, researchers interested in personalised liver SBRT using functional MRI should aim to answer the question—“What is the benefit of this approach to the patient?”. Functional MRI involves advanced image acquisition techniques, combined with complex mathematical modeling of the MRI signal to derive quantitative functional parameters. SBRT requires precise delineation of tumour volumes and accurate treatment delivery. Typically, margins to account for uncertainty in both are less than 5 mm. Current standard of care treatments for liver SBRT require multiple low dose CT scans to characterise tumour motion due to respiration, and if using breath hold techniques, it is necessary to confirm reproducibility of target position. Treatment volume margins are then applied to account for residual uncertainties. Functional MRI acquisition can take longer than a typical breath-hold duration. Advanced motion correction methods will need to be developed to align functional maps acquired in free breathing to the CT scans used for dose calculation and characterise tumour motion. These techniques are not currently implemented on MRI scanners and, therefore, will require offline image processing utilising considerable computational resources. Optimisation of liver SBRT planning based on spatial liver function distribution and mid-treatment plan adaptation is also non-trivial. Future studies must be performed not only to validate the methodology and establish the feasibility of the technique, but also provide robust evidence of real improvements in patient outcomes.

## 6. Conclusions

Liver SBRT is an attractive treatment option for patients with unresectable liver cancers, with promising outcomes demonstrated for patients with good overall liver function. In the case of patients with impaired baseline liver function, curative treatment options are limited. Functional MRI, however, offers the opportunity for safe delivery of personalised and adaptive liver SBRT using spatial liver function maps to optimise radiation dose distribution. Prospective, longitudinal studies using standardised MRI protocols and the development of dose–response models of liver function will enable patients with impaired overall function to also be offered curative treatment.

## Figures and Tables

**Figure 1 cancers-14-05860-f001:**
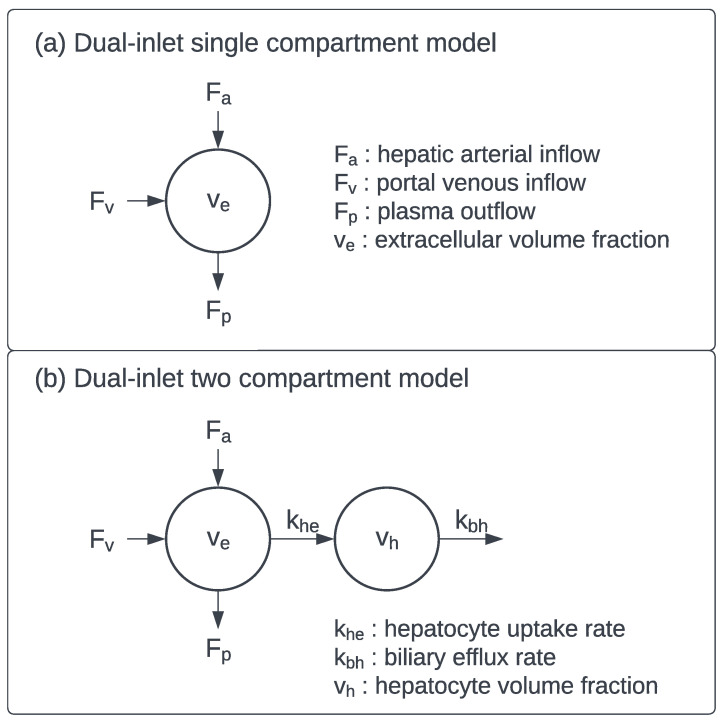
Tracer kinetic model diagrams for quantitative DCE-MRI in the liver parenchyma: (**a**) Dual-inlet single compartment model (**b**) Dual-inlet two compartment model.

**Table 1 cancers-14-05860-t001:** Diffusion models for quantitative DWI analysis. b: b-value; S: diffusion-weighted signal; S_0_: signal at b = 0 s/mm^2^.

Model	Model Equation	Parameters
Mono-exponential	$S=S_{0}e^{-b.\mathrm{ADC}}$	ADC: apparent diffusion coefficient
IVIM	$S=S_{0}\left({\mathrm{fe}}^{-b.D*}+\left(1-f\right)e^{-b.D}\right)$	D*: Pseudo-diffusion coefficient; f: perfusion fraction; D: diffusion coefficient
Kurtosis	$S=S_{0}e^{-b.D_{K}+\frac{b^{2}D_{K}^{2}K_{\mathrm{app}}}{6}}$	D_K_: diffusion coefficient; K_app_: diffusion kurtosis

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
