# Peer review of "Role of Functional MRI in Liver SBRT: Current Use and Future Directions"

_cancers, 2022, doi:10.3390/cancers14235860_

Round 1

Reviewer 1 Report

The manuscript under consideration presents a review on applications of fMRI diagnostics in conformal radiation therapy of liver tumors. The authors appropriately described advantages and disadvantages of different fMRI methods for SBRT treatment planning and response assessement in comparison with other techniques. Undoubltely the main advantage of fMRI is to visualize functional state of liver and nearby tissue with sufficient detail.

Perhaps, more information on potential difficulties could be given, for example, on computational expenses and signal post-processing, reproducibility of certain parameters, problems with organ movement in treatment planning, etc.

The manuscript is well written and structured, and describes promising approach to consider for radiotherapeuts society.

I would recommend to accept it for publication.

Author Response

We thank the reviewer for their feedback and appreciate their positive response. As suggested by the reviewer, we have added Lines 562-572 (Final paragraph of Discussion) in the revised manuscript describing the technical challenges relating to post processing, organ motion and increased computational requirements. 

Reviewer 2 Report

The review provides comprehensible information regarding the new techniques of SBRT used in liver cancer. The paper is well-written and organized. 

Indeed treatment planning requires high improvement toward personalized treatment for each patient. Are there any studies reporting longitudinal data for the use of MRI in the liver in order to validate the techniques presented? I would include a short description of the evidence regarding this matter (if exists) and the reasons why there is a lack of using these new techniques in clinical practice. You talked about mathematical modeling in cancer and I found it necessary to include recent works in this field (doi: 10.33909/jcm11041006, https://doi.org/10.1016/j.ifacol.2020.12.352).

How do you see the implementation of recent research for optimization of SBRT planning in the clinical practice, so the patients could benefit from it? The benefit is real and easy to be seen, but how optimistic are you about the use of this approach by the doctors/MRI companies? I find it valuable to explain how do you see the transfer of research results into clinical use.

Author Response

We thank the reviewer for their feedback. Please see our response to specific comments below. 

  • Are there any studies reporting longitudinal data for the use of MRI in the liver in order to validate the techniques presented? I would include a short description of the evidence regarding this matter (if exists) and the reasons why there is a lack of using these new techniques in clinical practice. 

The reviewer is correct, there are studies reporting longitudinal data on the MRI parameters in the liver. However, these studies are typically in the context of response to chemotherapy or chemo and radio-embolisation treatments such as TACE or SIRT, and only look at response of the tumour, not the surrounding liver tissue. There aren't any large datasets specifically looking at the radiation dose response of the functional MRI parameters in the context of external beam radiation therapy. We have added this information in Lines 538-542 (Discussion) to clarify the lack of longitudinal data on radiation dose response of liver tumour and non-tumour cirrhotic liver.

  • You talked about mathematical modeling in cancer and I found it necessary to include recent works in this field (doi: 10.33909/jcm11041006 , https://doi.org/10.1016/j.ifacol.2020.12.352).

Thank you for pointing us to these references. We have added them in Lines 492-493 (Section 4.3). 

  • How do you see the implementation of recent research for optimization of SBRT planning in the clinical practice, so the patients could benefit from it? The benefit is real and easy to be seen, but how optimistic are you about the use of this approach by the doctors/MRI companies? I find it valuable to explain how do you see the transfer of research results into clinical use.

Thank you for this comment. Indeed, the final goal must be a clinical benefit from the novel methods. We have added details on clinical translation in the Discussion in Lines 562-572 and also in the Conclusion section.